# EIMECA: A Proposal for a Model of Environmental Collective Action

Beatriz Carmona-Moya [ID], Antonia Calvo-Salguero [ID] and María-del-Carmen Aguilar-Luzón *[ID]

Department of Social Psychology, University of Granada, 18071 Granada, Spain; bcarmona@ugr.es (B.C.-M.); acalvo@ugr.es (A.C.-S.)
* Correspondence: maguilarluzon@ugr.es

**Abstract:** The deterioration and destruction of the environment is becoming more and more considerable and greater efforts are needed to stop it. To accomplish this feat, all members of society must identify with solving environmental problems, environmental collective action being one of the most relevant means of doing so. From this perspective, the analysis of the psychosocial factors that lead to participation in environmental collective action emerges as a priority objective in the research agenda. Thus, the aim of this study is to examine the role of "environmental identity", as conceptualized by Clayton, as a central axis for explaining environmental collective action. The inclusion of the latter in the theoretical framework of the SIMCA (social identity model of collective action) model gives rise to the model that we have called EIMECA (environmental identity model of environmental collective action). Two studies were conducted (344 and 720 participants, respectively), and structural equation modeling was used. The results reveal that environmental identity and a variety of negative emotional affects, as well as group efficacy, accompanied by hope for a simultaneous additive effect, are critical when it comes to predicting environmental collective action.

**Keywords:** environmental identity; environmental collective action; emotions; moral conviction; group efficacy beliefs



## 1. Introduction

Global warming, environmental pollution, forest destruction, soil degradation, water scarcity, or species extinction are, among many others, examples of the various problems that currently plague Mother Nature, and with her, plague humanity. Although there are many people who are not yet aware of this issue (or do not want to acknowledge it), it is also clear that there are many of us who can see that we are facing a great environmental crisis, and it is clear that protecting the environment is necessary and fundamental for the existence and preservation of both our planet and the human beings that inhabit it.

The magnitude of the current problem is such that, in recent times, the defense of the environment is one of the reasons why citizens have become engaged in social mobilization. These actions have not been in vain, since the report published in 2018 by Mexico's National Institute of Statistics and Geography (INEGI), called "Statistics for World Environment Day (5 June)" [1], states that collective actions in favor of the protection of the environment and natural resources are a reflection of the need for society to maintain a positive relationship with the environment.

Although science has shown that environmental degradation has its origin fundamentally in human behavior, i.e., that its causes are anthropogenic, there are still people who evince doubts about it, e.g., [2–4]. This skepticism damages the perception of shared or consensual beliefs and attitudes, which are relevant determining factors when it comes to carrying out pro-environmental behaviors that allow us to halt this deterioration [5–7]. Therefore, the way to curb environmental problems comes fundamentally from changing beliefs and attitudes in this regard, and consequently, from changing the behaviors that cause them. The relevance of environmental collective action in this context lies in the fact

that it is focused precisely on provoking a transformation or social change in beliefs and attitudes related to environmental protection, and generating shared or consensual beliefs and norms [8–10].

Consequently, given the relationship established in the literature between human behavior and environmental crises, we are currently witnessing the emergence of a research agenda committed to analyzing the social factors that trigger this relationship [11], with the factors that lead to collective action occupying a fundamental position.

The field of psychology plays a crucial role in identifying and explaining the factors that facilitate people's involvement in collective actions. These are behaviors carried out in a group—either directly or as a representative of an organization—that seek to satisfy the shared and perceived interests of the members of that group, with the aim of provoking social transformation or change [8–10]. Thus, this type of action differs from pro-environmental behavior in the private sphere or at the individual level, such as energy-saving, using public transport, lowering household consumption, or recycling [12].

Although various psychosocial factors have been identified by social and environmental psychology for the prediction of environmental collective action, social identity has emerged as a key factor [8,13–15], occupying a central role in the various theoretical models that have been put forward in this regard in recent decades [15–20]. On the one hand, following a review of the literature, we concluded that there is still only a relatively small number of studies that address the relationship between social identity and environmental collective action, and there is also little integration of the factors involved in the proposed models. This research area, therefore, still lacks a unified theoretical framework [8,12,14,15,21].

On the other hand, many of the models have been considered from a social psychology perspective, in the context of competitive collective action [22], that is, in the context of collective protest to reduce injustices and the structural disadvantages in society that are faced by low-status or disadvantaged groups [17–20]. Theoretically, these models can be transferred to the field of collective action by conversion [22], which characterizes environmental collective action. Although there have been some attempts to provide evidence of the latter [8], this remains a question that needs to be confirmed by much more research [8,15].

In addition, the few existing studies in the literature have focused on analyzing the role of group identity and, above all, politicized identity [8,16,21]. However, it is important to consider that within the domain of environmental behavior, the construct of "environmental identity", as proposed by Clayton [23], has emerged, which has been shown in several studies to have a positive and significant correlation with environmental collective action [13,24,25]. Consequently, we strongly believe that analysis of the role of this conceptualization of environmental identity must be addressed and integrated into the analysis of collective action models, in the specific context of environmental behavior.

In this paper, two studies are conducted in which we aim to address at least some of the limitations of previous studies, and also to contribute to the theoretical and empirical knowledge within the field of environmental collective action. Our original objective was to test the role of environmental identity in predicting these actions. To achieve this objective, we will take as a theoretical frame of reference one of the most relevant models on collective action that has been successfully tested in the context of socio-structural injustices, that is, SIMCA (social identity model of collective action) [19,20]. Likewise, we will take the conceptualization of the "environmental identity" posited by Clayton [23], which is both the most advanced and also the one that most closely resembles the concept of social-collective identity, from all of those existing in the environmental field [12,26,27]. The objective of the first study was to analyze the propositions of the SIMCA model using Clayton's environmental identity as the central axis, instead of the politicized social identity proposed by van Zomeren et al. [19,20]. Given that the results obtained in this first study were not fully satisfactory, a second study was carried out in which the conceptualization of two of the variables proposed by the SIMCA model was improved, taking into account

the results and suggestions found in the empirical literature. For this reason, we present the SIMCA model below, in order to later present the conceptualization of environmental identity from Clayton's perspective [23].

## 2. Social Identity Model of Collective Action (SIMCA)

This model describes the background of collective action carried out by both disadvantaged groups [19] and by favored or advantaged groups [28], to condemn situations of inequality and to promote social change. Although this model has been successfully tested in this context, with the exception of certain attempts [8,29], we have not been able to find further empirical evidence to conclude that the model works well within the context of environmental collective action. Furthermore, these studies have considered only a group identity or politicized identity, rather than the broader construct of environmental identity (see below).

Based on the previous literature, SIMCA proposes that group feelings based on anger or an emotional experience of injustice over collective disadvantage, and the perception of group efficacy and group identity, directly predict collective action. The main axis of SIMCA is identification with the disadvantaged group, considering that the motivations for changing social inequality require a strongly developed social identity [19,20]. Thus, individuals who identify more strongly with the group, as opposed to those who identify less, are more committed to the situation, goals or objectives and group interests, and pay more attention to shared group norms concerning the actions required to achieve such goals. The authors distinguish between group identity and politicized identity. Politicized identity implies identification with a social movement or organization, and that takes responsibility for the interests of the group [30]. Therefore, the authors propose and confirm in their study [20] that a politicized identity, but not a non-politicized identity, allows for predicting the collective action, since the former is more normatively oriented than the latter toward such action, and the members of the group feel a stronger internal obligation to participate in the activities of an organization of the social movement [19,31]. Further, the model also assumes that politicized identification not only directly predicts collective action, but also indirectly, because it increases group feelings based on anger, while at the same time increasing the perception of group efficacy [32,33]. However, the authors propose that moral convictions are at the root of politicized identification, the emotional experiences of anger and injustice, and the sense of group efficacy [20,28].

Taking the assumptions of this model as a reference framework, in this paper we integrate Clayton's conceptualization of "environmental identity" [23] into this theoretical framework. This is regarded as the central axis for predicting environmental collective action. We have given the resulting model the acronym EIMECA (environmental identity model of environmental collective action). Before continuing to present the model, it is necessary to define even more precisely how environmental identity is going to be conceptualized in the present paper.

## 3. Conceptualization of Environmental Identity in the EIMECA Model

The existing models that focus on social identity as a means of explaining collective action are based on the approaches of the social identity theory and the self-categorization theory [34–37]. According to the first approach, a social identity reflects a collective identity, i.e., the process by which people identify with a social category or a collective, such as a group, which leads the group to mobilize toward collective action. From that approach, as suggested above, we believe that the construct of "environmental identity", as conceptualized by Clayton (2003), deserves attention in order to predict environmental collective action.

In this context, it is important to consider that in the field of environmental behavior other identity constructs such as "green personal self-identity", "role identity", "place identity" or "place attachment", and "environmental identity" (for more details, see [15]) have developed or emerged, which, in the opinion of some authors, should not be confused

with social-collective identity, since they do not refer completely and distinctively to the processes of identification at the group level, nor to the collective self-definitions that are of interest to explain collective action in the face of the environmental crisis [15]. Along the same lines, Tam [27] empirically analyzes the differences and similarities between constructs, such as "commitment to nature", "connectedness to nature", "connectivity with nature", "emotional affinity toward nature", "environmental identity", "inclusion of nature in self", and "nature relatedness" (for more details, see [27]), and concludes that the constructs related to the connection with nature that are multidimensional, such as that of Clayton [23], capture in good part the dimensions of relevant models of "social-collective identity" such as the Ashmore et al. [38] model (self-categorization, importance, evaluation, attachment and sense of interdependence, and behavioral participation) and the Cameron model [39] (the three dimensions of this model are cognitive centrality, intragroup affect and intragroup bonds). It should be noted here that, taking as a point of reference the definition of social identity proposed by Tajfel [40], Ellemers et al. [41] distinguish three fundamental elements of social identification: the cognitive (the individual's knowledge of group membership, or self-categorization), the evaluative (positive or negative value linked to group membership, or group self-esteem) and the emotional (sense of emotional involvement with the group, or affective commitment).

In general, the results of other studies provide empirical evidence for the ideas suggested by Tam [27] regarding the elements of the content underlying the environmental identity, as well as for the comments of other authors who have also suggested that this scale refers to ideas related to "social-collective identity" [12,26,27]. For all these reasons, as indicated above, we believe that Clayton's conceptualization of environmental identity emerges as a conceptualization of social identity, which is of interest for the explanation of collective action in the environmental context, and that it should be addressed in the theoretical framework of models in this regard.

According to Clayton [23], nature can be conceived, just as social groups are conceived, as a community or collective, not exclusively human, but to which human beings belong. Therefore, it becomes possible to speak of a connection between nature (as a collective) and people, which affects the way they perceive it, and which becomes an important part of their own self-concept. This connection with nature was defined by Clayton as "environmental identity", and, based on this conceptualization, a scale was created with which to measure it, based on theories about the factors that determine a "social-collective identity" [40,42,43]. The scale comprises multiple dimensions representative of the following factors: interaction with nature (the prominence of identity in the group), the importance of belonging to nature (the identification of oneself as a member of the group), the importance of nature (the agreement with an ideology associated with the group) and positive emotions toward nature (the positive emotions associated with the group). (See examples of items in Section 6: Methods).

## 4. EIMECA Model (Environmental Identity Model of Environmental Collective Action)

The EIMECA model is the result of integrating environmental identity, within the theoretical framework of the SIMCA model, as the central axis for predicting environmental collective action. It should be noted that here, unlike the SIMCA model, we are not going to enter into a discussion of the distinction between politicized or non-politicized identity. Theoretically [17,44–49], both can predict collective action, and there are studies, even in the field of environmental collective action [8,12,19,29], in which the predictive capacity of identity with the group or non-politicized identity is revealed. Our central interest is in the predictive capacity of environmental identity, as conceptualized by Clayton [23].

To understand the links between environmental identity and the rest of the variables proposed by the SIMCA model, it must be taken into account, as presented above, that this is theoretically comparable with the conceptualization of the social-collective identity of the model [19,20] since both share the same theoretical approach to the construction of social identity [35,36]. From this perspective, and according to the assumptions of the SIMCA

model, the relationships proposed in the EIMECA model should also be confirmed for the concept of environmental identity. Therefore, it is to be expected, at least theoretically, that the EIMECA model will fit well, that its predictive capacity will be adequate, and that environmental identity will prove to be the central axis of the model.

Finally, it should be noted that previous research has used a variety of measures of environmental collective action (e.g., [8,12,16,20,21,28]), including those from measures of behavioral intention to measures of actual behavior, as well as very varied items in terms of the type of behavior or action to be carried out. Even so, these investigations have revealed the predictive capacity of different antecedent factors regarding the diversity of collective action measures used. However, most of these investigations have not taken into account the different levels of participation or personal involvement in this type of action. In this sense, Alisat and Riemer [24] emphasize the need to distinguish between participatory actions and leadership actions. According to the authors, the difference between the two types of actions would be determined by the degree of social and political pressure that they entail. Participatory actions carried out by citizens, such as keeping informed about an environmental issue, and/or holding conversations with other people on these issues, include simple behaviors that do not require high involvement and, furthermore, entail little social pressure or politics. While leadership actions such as, for example, organizing events and groups for the defense of the environment, e.g., organizing a mobilization, a boycott or a citizen protest, are more complex, they require greater personal involvement on the part of the person, necessitating that they possess the competences, resources and personal abilities that these actions imply and, furthermore, that they have a more socio-political nature. In the present study, given that there are theoretical and empirical foundations to distinguish between both types of environmental collective actions, they will be considered in the testing of the model. In line with what has been stated so far, it is expected that the variables proposed by the model will be capable of predicting both types of actions. In order to test the EIMECA model, two studies were carried out, which are described below.

## 5. Study 1

The aim of the first study was to test the EIMECA model, with the main purpose of verifying the role of environmental identity in predicting environmental collective action. Given that the assumptions proposed by the EIMECA model have already been set out above, we invite the reader to look at the theoretical framework of the model in Figure 1.

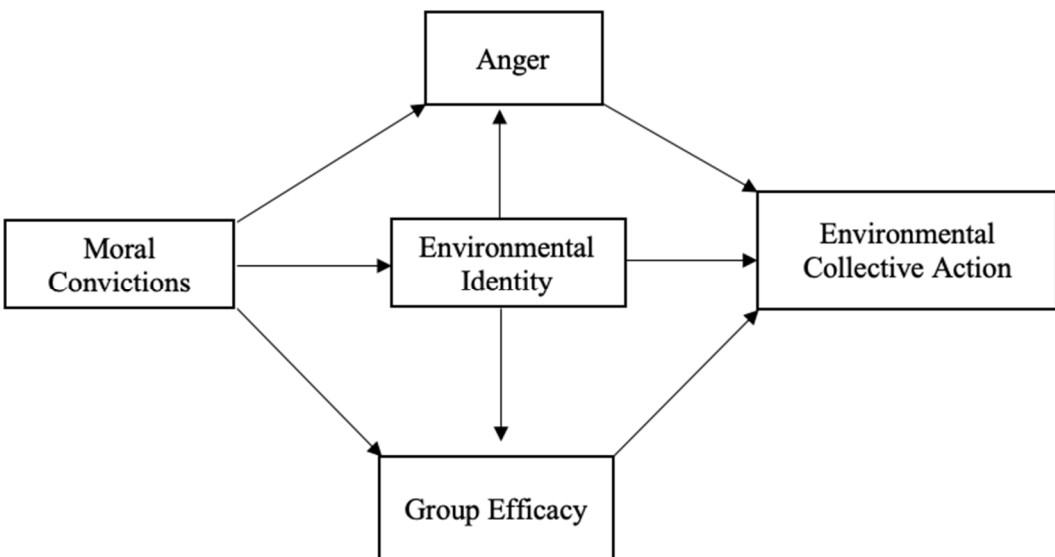

**Figure 1.** Theoretical framework of the EIMECA (environmental identity model of environmental collective action).

**Hypothesis 1a.** *Moral convictions directly predict anger, group efficacy beliefs and environmental identity.*

**Hypothesis 1b.** *Environmental identity predicts anger and group efficacy beliefs.*

**Hypothesis 1c.** *Anger, group efficiency beliefs and environmental identity directly predict environmental collective action measures.*

**Hypothesis 1d.** *The effect of environmental identity on environmental collective action measures is greater than that of the other variables in the model.*

## 6. Methods

### 6.1. Participants and Procedure

The sample was composed of a total of 344 participants (general population), of which 27.6% were men, and 72.4%, women. The participants were of Spanish nationality, with an average age of 24.59 years (SD = 8.16). The data were collected after obtaining approval by the Ethics Committee of the University of Granada. All participants read the instructions for participation in the study, and were assured that their answers would remain confidential. The participants were not required to provide any personal data that could identify them. Given the advantages of online data collection [50–54], this method was chosen to distribute and complete the questionnaire. The questionnaire was created through the Limesurvey platform provided by the University of Granada, and was later disseminated through various social networks (Facebook, Twitter, WhatsApp and Instagram). We decided to use these media, since social networks present unique opportunities for rapid and cost-effective data collection from populations with very specific demographics or interests [55]. The responses were entered directly into spreadsheets that were then imported into statistical software (SPSS). In addition, AMOS version 24 was used to assess hypothesized relationships, as well as the degree of model fit.

### 6.2. Variables and Measuring Instruments

Moral convictions about environmental protection: We used six items from the study by van Zomeren et al. [28], adapted to the environmental context. An example of an item is: "My opinion about environmental degradation is an important part of my moral norms and values". Participants were asked to respond to each item on a 5-point Likert-type scale, from (1), indicating "not at all", to (5), indicating "strongly agree".

Anger: Defined as a negative feeling or emotion of rage or wrath [56], this was measured using the following items, adapted to the environmental context, from the study by Shepherd et al. [57]: (1) "To what extent do you feel annoyed about the measures taken to alleviate the effects of environmental degradation?"; (2) "To what extent do you feel angry about the measures carried out to alleviate the effects of environmental deterioration?"; and (3) "To what extent do you feel indignant about the measures carried out to alleviate the effects of environmental deterioration?". These are evaluated on a 5-point Likert-type scale (1 = strongly disagree, 5 = strongly agree).

Environmental Identity: This was measured using Clayton's [23] environmental identity scale (EID), adapted to the Spanish context by Olivos and Aragonés [58]. This scale is composed of 24 items, with a 5-point Likert response scale (1: very much in disagreement; 5: very much in agreement) that evaluates four dimensions of environmental identity: "enjoying nature", "appreciation of nature", "environmental identity" and "environmentalism". Examples of the items are: "I think of myself as part of nature, not separate from it"; "I have a lot in common with environmentalists".

Group Efficacy Beliefs: Participants were required to express their degree of agreement (from 1, "not at all", to 5, "strongly agree") with 4 items used in the study by van Zomeren et al. [20], which were adapted to environmental behavior. An example of an

item is: "As inhabitants of this planet, I think we can successfully defend our natural resources together".

Environmental Collective Action: The environmental collective action scale (EAS) [24], adapted to the Spanish context by Carmona-Moya et al. [25], was used. The question participants were required to think about when answering is: "In the last six months, how often have you participated in the following environmental activities or actions?" These activities are evaluated through 16 items in a 5-point Likert-type response format, where (0) is "never" and (4) is "frequently". This scale provides a score both globally and for two distinct dimensions: leadership actions (LA) and participation actions (PA). An example of an item from the participation dimension is: "I have participated in a community event focused on raising environmental awareness (such as cleaning beaches, forests, etc.)". An example of an item from the leadership dimension is: "I have taken part in a protest or demonstration about an environmental issue".

## 7. Results

First, a descriptive analysis of the variables was carried out, whilst Cronbach's alpha value was calculated for each scale. The mean scores of all variables were relatively high (above the scale mean). Pearson's correlation analyses were then conducted (see Table 1). The results of these analyses revealed significant correlations between the different variables.

**Table 1.** Descriptive and reliability analysis, and correlations between key measures.

|  | M(SD) | Cronbach's Alpha | 1 | 2 | 3 | 4 | 5 | 6 | 7 | 8 | 9 |
|---|---|---|---|---|---|---|---|---|---|---|---|
| 1. MC | 3.93 (0.90) | 0.87 | - | 0.340 ** | 0.353 ** | 0.558 ** | 0.336 ** | 0.401 ** | 0.152 ** | 0.004 | −0.099 |
| 2. ANGER | 3.65 (1.11) | 0.90 |  | - | 0.198 ** | 0.340 ** | 0.206 ** | 0.239 ** | 0.106 * | −0.021 | −0.145 ** |
| 3. GEB | 4.49 (0.71) | 0.90 |  |  | - | 0.397 ** | 0.037 | 0.103 | −0.095 | −0.003 | −0.084 |
| 4. EID | 3.71 (0.71) | 0.94 |  |  |  | - | 0.441** | 0.509 ** | 0.231 ** | 0.059 | −0.054 |
| 5. EAS_GL | 2.03 (0.73) | 0.92 |  |  |  |  | - | 0.969 ** | 0.881 ** | 0.091 | 0.011 |
| 6. EAS_PA | 2.34 (0.82) | 0.88 |  |  |  |  |  | - | 0.736 ** | 0.073 | −0.037 |
| 7. EAS_LA | 1.50 (0.71) | 0.82 |  |  |  |  |  |  | - | 0.110 * | 0.100 |
| 8. AGE | 24.59 (8.16) | - |  |  |  |  |  |  |  | - | 0.023 |
| 9. GENDER [a] | - | - |  |  |  |  |  |  |  |  | - |

MC = moral conviction; GEB = group efficacy belief; EID = environmental identity; EAS_PA = environmental action scale—participation actions; EAS_LA = environmental action scale—leadership actions; EAS_GL = environmental action scale—global. [a] Female: 1; Male: 2; * correlation is significant at the 0.05 level; ** correlation is significant at the 0.01 level.

*Predicting Environmental Collective Action: Structural Equation Modelling (Path Analysis)*

In order to test the hypotheses of the EIMECA model, as well as its fit to the data, structural equation analyses (path analyses) were carried out using the AMOS version 24 statistical package. Given the condition of multivariate normality presented by the variables of the study, the maximum likelihood estimate [59] was used. Since, in testing the model, AMOS suggested a slightly better fit with the inclusion of the direct effects of moral convictions on environmental collective action measures, the analyses were repeated, obtaining the following estimates for global collective action: CMIN/DF = 0.569; CFI = 1.000; TLI = 1.013; RFI = 0.984; NFI = 0.998; SRMR = 0.0089; RMSEA = 0.000; and for global collective participation: CMIN/DF = 0.569; CFI = 1.000; TLI = 1.012; RFI = 0.985; NFI = 0.998; SRMR = 0.0089; RMSEA = 0.000. For collective leadership action, AMOS did not suggest including any extra relationships to those already established in the model: CMIN/DF = 0.895; CFI = 1.000; TLI = 1.004; RFI = 0.969; NFI = 0.994; SRMR = 0.0148; RMSEA = 0.000. The estimates of the standardized coefficients found for the different model paths, together with their significance, are displayed in Figures 2–4. The percentage variance in global collective action measure explained by the model was 23.8%; for participation, this was 30%, and for leadership, 9.7%. Environmental identity explained 31.1% of the variance in each of the three environmental collective action measures.

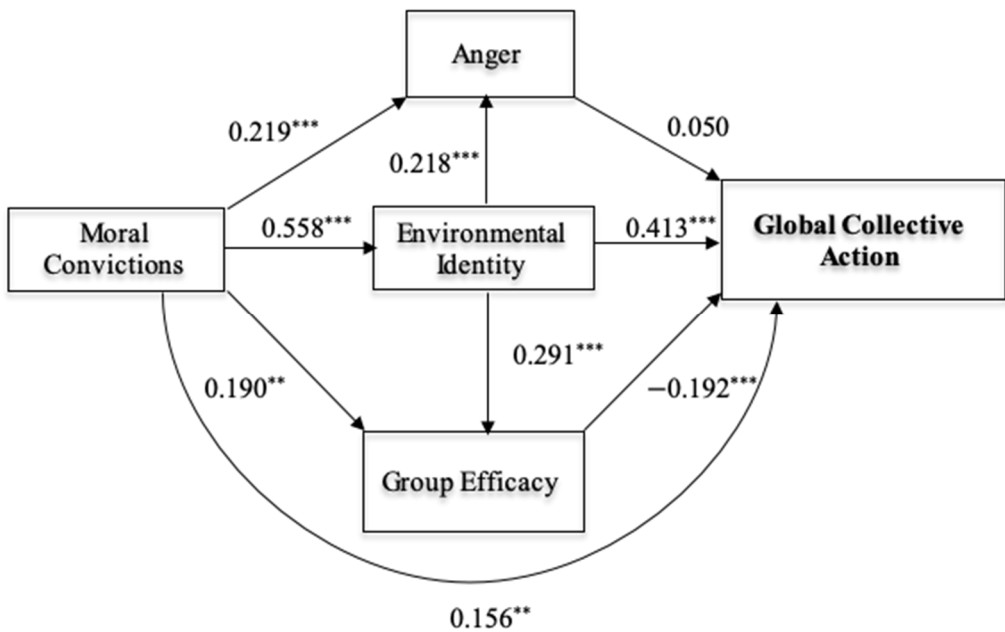

**Figure 2.** Estimates of the standardized coefficients for the different EIMECA model paths: global collective action.
** $p < 0.01$. *** $p < 0.001$.

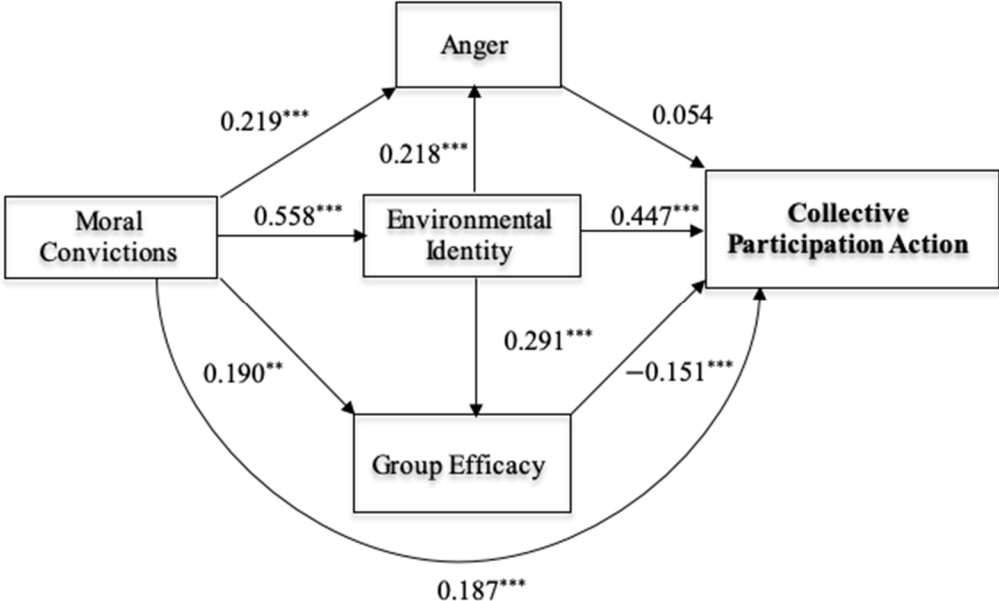

**Figure 3.** Estimates of the standardized coefficients for the different EIMECA model paths: collective participation action.
** $p < 0.01$. *** $p < 0.001$.

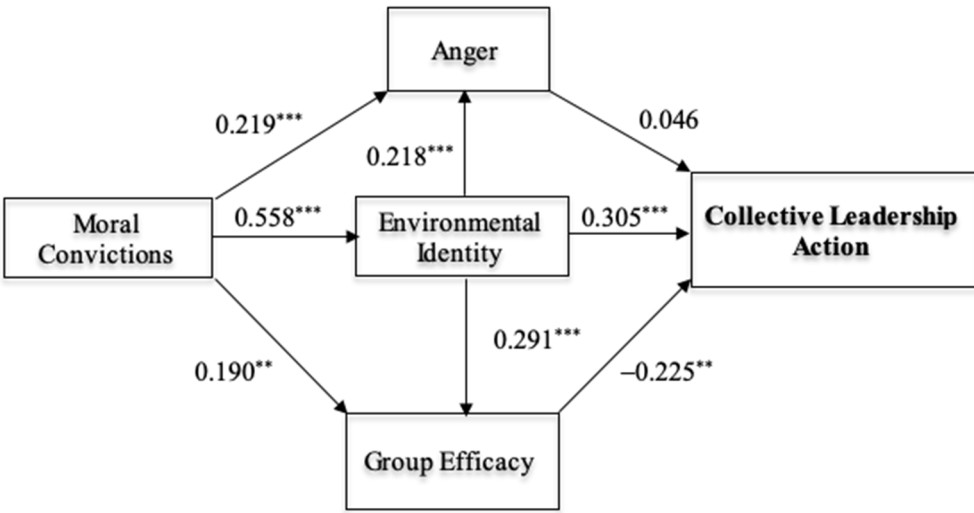

**Figure 4.** Estimates of the standardized coefficients for the different EIMECA model paths: collective leadership action.
** *p* < 0.01. *** *p* < 0.001.

### 8. Brief Discussion (Study 1)

The results of Study 1 confirmed Hypothesis 1a (moral convictions directly predict anger, group efficacy beliefs, environmental identity) and Hypothesis 1b (environmental identity predicts anger and group efficacy beliefs). With regard to Hypothesis 1c, it was confirmed that environmental identity directly predicts the three actions, but the predictive capacity of anger was not confirmed and, although a significant relationship was obtained, the positive effects of group efficacy were also not confirmed, since they were negative. These latter results are in line with those reported in the study by Bamberg et al. [8], since no relationship was found between the negative emotions of anger, indignation and rage, and environmental collective action intent, or between group efficacy beliefs and such behavioral intent.

With regard to the degree of fit of the model, the estimates of the different indicators were excellent [60–62]. In this aspect of the model's results, it should be borne in mind that a new path was included that represented a direct effect of moral convictions on environmental collective action measures. This effect can be justified theoretically, particularly if we consider that moral convictions are experienced as strong and absolute positions that do not acknowledge exceptions to the higher-order principle. Thus, the costs associated with not acting in a manner consistent with what is believed [63,64], and the need to reaffirm the moral stance, lead to the need to act, as these are placed at a higher level of importance than any of the various identities that one may have [28]. Thus, participation in collective actions represents behavior that is morally consistent with those moral convictions.

Furthermore, the model explains a relatively moderate percentage of the variance of the three actions. This suggests that there must be other fundamental variables that explain these actions. Other studies have shown the importance of variables such as the personal norm [65,66] and group norms [67,68], and perceived behavioral control [8] or moral obligation [16]. Even though the model could lose parsimony, future research could include these variables in the model. Likewise, the influence of other socio-structural and political factors that make up the context in which these actions take place must be taken into account, in addition to the psychological factors contemplated by the model [19]. However, it should be noted that environmental identity explained the highest percentage of this variance (31.1%), and, in comparison with the remaining variables, it obtained the highest coefficient in its relationship with environmental collective action measures, becoming the main variable in the model and, consequently, confirming the central hypothesis of our study (Hypothesis 1d).

Despite these encouraging results, the absence of a significant relationship between anger and environmental collective action measures is of concern, as is the significant but negative relationship between group efficacy beliefs and actions. There are several reasons for these results, including the conceptualization and operationalization of the two variables in the specific context of environmental collective action. Therefore, it is undoubtedly the case that these first results can be improved. For this reason, a second study was carried out.

## 9. Study 2

The objective of this study was to obtain full support for the relationships established in the EIMECA model, whilst overcoming some of the limitations of Study 1, by improving the conceptualization and operationalization of perceived group efficacy beliefs and ameliorating the negative effects derived from the perception of environmental deterioration and environmental problems. Although the expected effect of group efficacy beliefs on collective action is well documented in several studies [19] there are also other works where this effect has not been found [8]. In view of this situation, it appears that there are studies showing that although people voice interest in problems such as climate change and other environmental threats, they still experience feelings of hopelessness, pessimism and helplessness, as well as inactivity [69–74]. Pessimism appears to be particularly strong when it comes to environmental issues [75]. Therefore, it is possible to suggest that the negative relationship found in our first study is due to the effect that certain emotions, such as hopelessness, could have on that relationship. In support of this suggestion, Cohen-Chena and van Zomeren [76] propose and confirm in their study that such beliefs of group efficacy only motivate collective action when hope is high, but not when hope is low. Therefore, the second study set out to test whether the interactive effects of group efficacy beliefs with the emotion of hope better predict collective action when compared with group efficacy beliefs alone.

Furthermore, in this second study, we also took into account the suggestions of some authors regarding the negative affects derived from the perception of deterioration and environmental problems. In this sense, several authors have proposed that, unlike competitive collective action, collective action by conversion [22]—as is the case with environmental collective action—may require the intervention of other negative emotions that are more relevant than anger, such as guilt or shame, since anger is the result of the evaluation of the behavior of majority groups or those in power, while in environmental collective action, negative emotions may also be the result of self-evaluation of behavior [29,77,78]. On the other hand, Kollmuss and Agyeman [79] argue that the greater the emotional involvement of people in evaluating the state of the environment, the greater the level of commitment to generate more pro-environmental actions. Therefore, we believe that measuring a wider range of negative emotions, rather than just anger, could improve the prediction of environmental collective action.

In this second study the hypotheses tested were:

**Hypothesis 2a.** *Moral convictions directly predict negative affects, group efficacy beliefs* X *hope and environmental identity.*

**Hypothesis 2b.** *Environmental identity predicts negative affects and group efficacy beliefs* X *hope.*

**Hypothesis 2c.** *Negative affects, group efficiency beliefs* X *hope, and environmental identity directly predict environmental collective action measures.*

**Hypothesis 2d.** *The effect of environmental identity on environmental collective action measures is greater than that of the other variables in the model.*

## 10. Method

### 10.1. Participants and Procedure

The sample in this study was composed of 720 participants (general population), of whom 31.5% (*n* = 227) were men and 68.5% were women (*n* = 493). The participants were of Spanish nationality, with an average age of 28.56 years (SD = 11.90). The procedure used for data collection was the same as that described for Study 1.

### 10.2. Variables and Measuring Instruments

Moral beliefs about environmental protection, perceived group efficacy, environmental identity and environmental collective action measures were assessed using the same scales as in Study 1.

The negative affective states derived from the perception of environmental deterioration were evaluated by means of the 10 items of the PANAS [80] negative affect scale, adapted to the Spanish context by López-Gómez et al. [81]. Responses to each emotional state were assessed on a 5-point Likert scale (from 1 = "slightly or not at all", to 5 = "very much"). The evaluation of each effect was adapted to the context of environmental deterioration, presenting the items, for example, as follows: "Thinking about the last month, how much GUILT have you felt about environmental degradation?". Another example is: "Thinking about the last month, how ASHAMED have you felt about environmental degradation?".

With respect to the variable, hope, the same five items from the Cohen-Chena and van Zomeren study [76] were used, adapted to the context of environmental problems. These items were accompanied by a 5-point Likert response scale (from 1 = "strongly disagree" to 5 = "strongly agree"). An example of an item is: "I feel hopeful about the possibility of solving the problem of environmental degradation".

## 11. Results

First, various descriptive, scale reliability and Pearson correlation analyses were carried out, the results of which can be seen in Table 2.

**Table 2.** Descriptive and reliability analyses, and correlations between the key measures of Study 2.

| | M(SD) | Cronbach's Alpha | 1 | 2 | 3 | 4 | 5 | 6 | 7 | 8 | 9 | 10 |
|---|---|---|---|---|---|---|---|---|---|---|---|---|
| 1. MC | 3.93 (0.85) | 0.87 | - | 0.459 ** | 0.300 ** | 0.303 ** | 0.561 ** | 0.342 ** | 0.387 ** | 0.220 ** | 0.128 ** | 0.045 |
| 2. NA | 2.91 (0.92) | 0.91 | | - | 0.176 ** | 0.294 ** | 0.512 ** | 0.355 ** | 0.387 ** | 0.255 ** | −0.108 ** | 0.195 ** |
| 3. GEB | 4.55 (0.65) | 0.93 | | | - | 0.040 | 0.323 ** | 0.002 | 0.050 | −0.082 * | −0.008 | 0.049 |
| 4. HOPE | 2.28 (0.94) | 0.85 | | | | - | 0.314 ** | 0.375 ** | 0.350 ** | 0.370 ** | 0.290 ** | −0.089 * |
| 5. EID | 3.70 (0.72) | 0.94 | | | | | - | 0.455 ** | 0.510 ** | 0.302 ** | 0.176 ** | −0.012 |
| 6. EAS_GL | 1.72 (0.95) | 0.91 | | | | | | - | 0.973 ** | 0.918 ** | 0.154 ** | −0.069 |
| 7. EAS_PA | 2.02 (1.00) | 0.92 | | | | | | | - | 0.803 ** | 0.124 ** | −0.040 |
| 8. EAS_LA | 1.21 (0.97) | 0.87 | | | | | | | | - | 0.186 ** | −0.109 ** |
| 9. AGE | 28.56 (11.90) | - | | | | | | | | | - | −0.168 ** |
| 10. GENDER [a] | - | - | | | | | | | | | | - |

MC = moral conviction; NA = negative affects; GEB = group efficacy belief; EID = environmental identity; EAS_GL = environmental action scale—global; EAS_PA = environmental action scale—participation actions; EAS_LA = environmental action scale—leadership actions; [a] Female: 1; Male: 2; * correlation is significant at the 0.05 level; ** correlation is significant at the 0.01 level.

The results revealed significant relationships between the different variables, with the exception of the relationship between group efficacy beliefs and hope, and with environmental collective action measures. It is worth noting that group efficacy beliefs were negatively and significantly correlated with leadership collective action. It should also be noted that the mean scores of all the predictor variables were relatively high (at or above the mean of the scale), with the exception of hope, which is in line with the results of other studies [69–74] that also indicate a low degree of hope in relation to the resolution of environmental problems. On the other hand, it is worth highlighting the positive and significant relationship between negative affective states and hope. This relationship suggests that the negative emotions experienced from the perception of environmental deterioration may be a relevant factor in the generation of hope related to the halting of such deterioration. This result, therefore, can be considered as a possible way for the activation

of environmental collective action, given the significant correlations between hope and environmental collective actions found in this study. Finally, it is worth highlighting the correlations of the sociodemographic variables, according to which, age is related to the three measured of collective environmental action, although gender is only significantly related to the leadership dimension.

*Predicting Environmental Collective Action: Structural Equation Modelling (Path Analysis)*

The EIMECA model in this second study was tested using the AMOS version 24 statistical package. Structural equation models were created using path analysis. First, the model was tested by including the measure of group efficacy alone (without the moderating effect of hope). Given that, in testing the model, AMOS suggested a slightly better fit with the inclusion of the direct effects of moral convictions on environmental collective action measures, the analyses were repeated, obtaining the following fit indicators for global collective action: CMIN/DF = 0.427/1 = 0.427; CFI = 1.000; TLI = 1.07; RFI = 0.995; NFI = 0.999; SRMR = 0.0050; RMSEA = 0.000); and for participatory collective action: CMIN/DF = 0.427/1 = 0.427; CFI = 1.000; TLI = 1.006; RFI = 0.995; NFI = 1.000; SRMR = 0.0050; RMSEA = 0.000. For collective leadership action, AMOS did not suggest including any extra relationships to those already established in the model: CMIN/DF = 1.898; CFI = 0.998; TLI = 0.988; RFI = 0.974; NFI = 0.995; SRMR = 0.0139; RMSEA = 0.035.

Regarding the standardized coefficients found for the various paths of the model, it should be noted that all were significant and positive, including the positive effect of negative affects on environmental collective action measures, but with the exception of the negative effect of group efficacy beliefs on environmental collective action measures (these coefficients are available from the first author). The percentage of variance explained by the model for global collective action was 26%; for participation, it was 30.8%, and for leadership, 14%. Environmental identity explained 31.1% of the variance in each of the three environmental collective action measures.

Second, before testing the model for the effects of an interaction between group efficacy and hope, we verified whether these effects were indeed evident on the three environmental collective action measures. Moderation analyses were conducted through the PROCESS package [82], following the suggestions of its author. The Model 1 template was employed, controlling both environmental identity and negative affects. The results revealed no significant interaction effects for global collective action ($\beta = -0.0297$; $p = 0.544$), participatory collective action ($\beta = -0.0043$; $p = 0.933$), and leadership collective action ($\beta = -0.0719$; $p = 0.176$). Hierarchical block regression analyses revealed significant positive main effects of hope on the global collective action measure ($\beta = 0.229$; $p = 0.000$), participatory collective action measure ($\beta = 0.182$; $p = 0.000$), and leadership collective action measure ($\beta = 0.280$; $p = 0.000$). There were also significant negative main effects of group efficacy beliefs on the global collective action measure ($\beta = -0.148$; $p = 0.000$), participatory collective action measure ($\beta = -0.117$; $p = 0.000$), and leadership collective action measure ($\beta = -0.182$; $p = 0.000$). The estimates of conditional effects offered in the analyses with PROCESS, as well as their graphical representation, suggested that as participation in environmental collective action measures increased, the values of group efficacy beliefs and hope simultaneously increased. Since these results are inconsistent with the main effects obtained in the regression analyses, i.e., the negative effects of group efficacy and the positive effects of hope, this seems to indicate the possibility of joint rather than interactive additive effects. That is, only when group efficacy beliefs and hope are both high (or low) is there a positive effect (or not) on environmental collective action measures. Therefore, we proceeded to test such effects through the estimation of a new variable (which we called "hopeful group efficacy belief") that combined this condition, that is, high scores on both variables and low scores on both variables. For this purpose, the construction of a continuous additive variable was chosen in preference to a categorical variable. The following formula was applied to obtain this variable: "Group Efficacy Score + Hope Score − | Group Efficacy Score − Hope Score |". This formula avoids the possibility that when adding up the

scores, a low score on one variable is compensated by a high score on the other, and vice versa, matching high (low) scores on both variables. Hierarchical block regression analyses, controlling for negative affect and environmental identity, revealed significant effects of hopeful group efficacy belief on all three environmental collective action measures (global: $\beta$ = 0.235; $p$ = 0.000; participation: $\beta$ = 0.187; $p$ = 0.000; leadership: $\beta$ = 0.286; $p$ = 0.000).

The model for the three environmental collective action measures was then tested, and the new variable was included. Since AMOS suggested that the relationship between the new variable (expected group efficacy) and negative affects could slightly improve the fit of the model, this was tested, obtaining the following model fit indices: global collective action measure: CMIN/DF = 1.931/1 = 0.1.931; CFI = 0.999; TLI = 0.989; RFI = 0.978; NFI = 0.998; SRMR = 0.0091; RMSEA = 0.036; participatory collective action measure: CMIN/DF = 4.737/1 = 0.4.737; CFI = 0.996; TLI = 0.959; RFI = 0.948; NFI = 0.995; SRMR = 0.0138; RMSEA = 0.072; leadership collective action measure: CMIN/DF = 0.002/1 = 0.002; CFI = 1.000; TLI = 1.013; RFI = 1.000; NFI = 1.000; SRMR = 0.0003; RMSEA = 0.000.

The estimates of the standardized coefficients of each path, together with their significance for the model with the relationship of the new variable, hopeful group efficacy belief, and negative affects, can be observed in Figures 5–7. The percentage variance in the global collective action measure explained by the model was 27.5%, that of participation was 31.2%, and that of leadership was 17.7%. The percentage of explained variance of the expected group efficacy beliefs of each of the environmental collective action measures was 12.7%, and that of environmental identity was 31.5%.

Finally, given the significant correlations obtained by the sociodemographic variables (age and gender) in relation to the measures of environmental collective action, multiple linear regression analyses per block were carried out, in which the effects of both variables were controlled on environmental collective action measures. The predictor variables of the model were taken as independent variables. The results obtained revealed that when the effects of these sociodemographic variables are controlled, all the predictor variables of the model significantly predicted the three measures of environmental collective action (data and results are available from the first author). These results support the results found through path analysis.

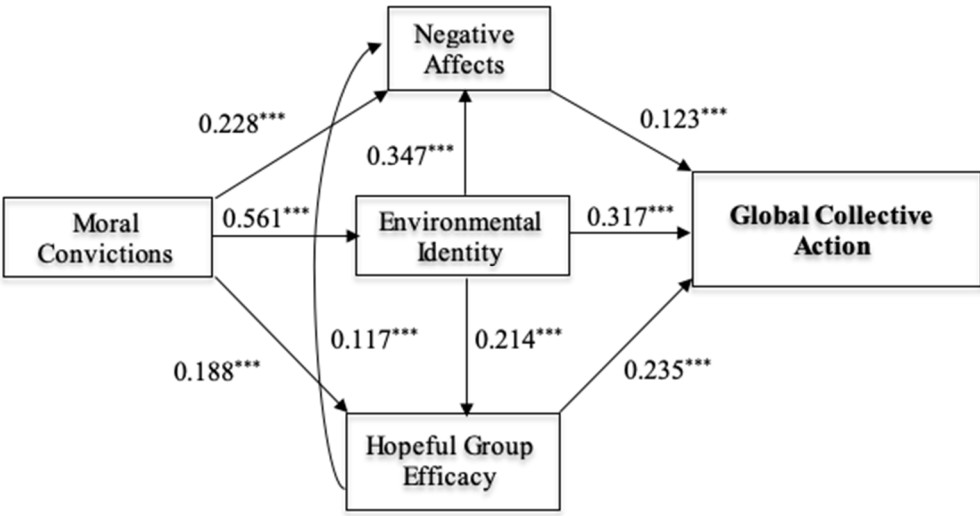

**Figure 5.** Estimates of the standardized coefficients found for the different relationship paths of the EIMECA model, including hopeful group efficacy instead of group efficacy belief: global collective action. *** $p < 0.001$.

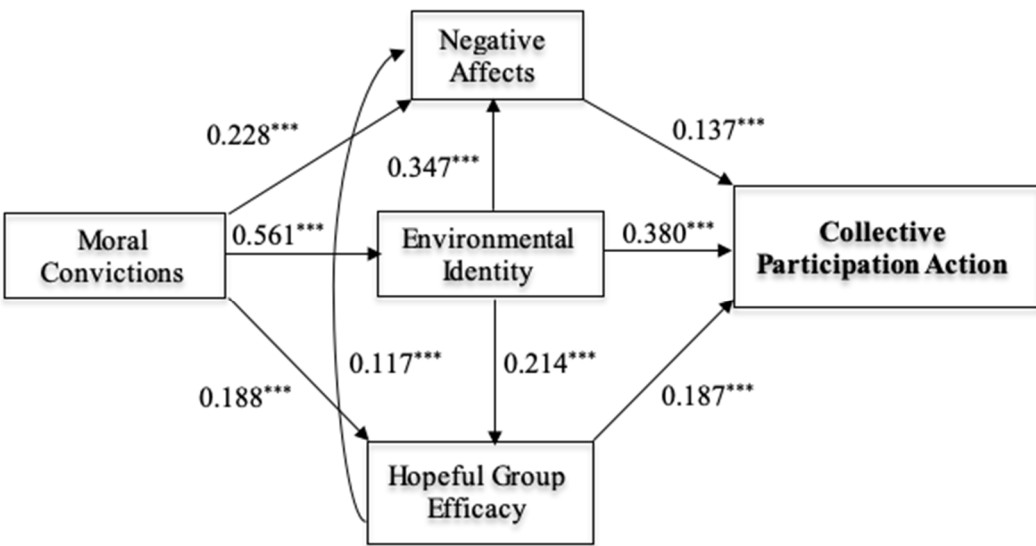

**Figure 6.** Estimates of the standardized coefficients found for the different relationships. *** $p < 0.001$.

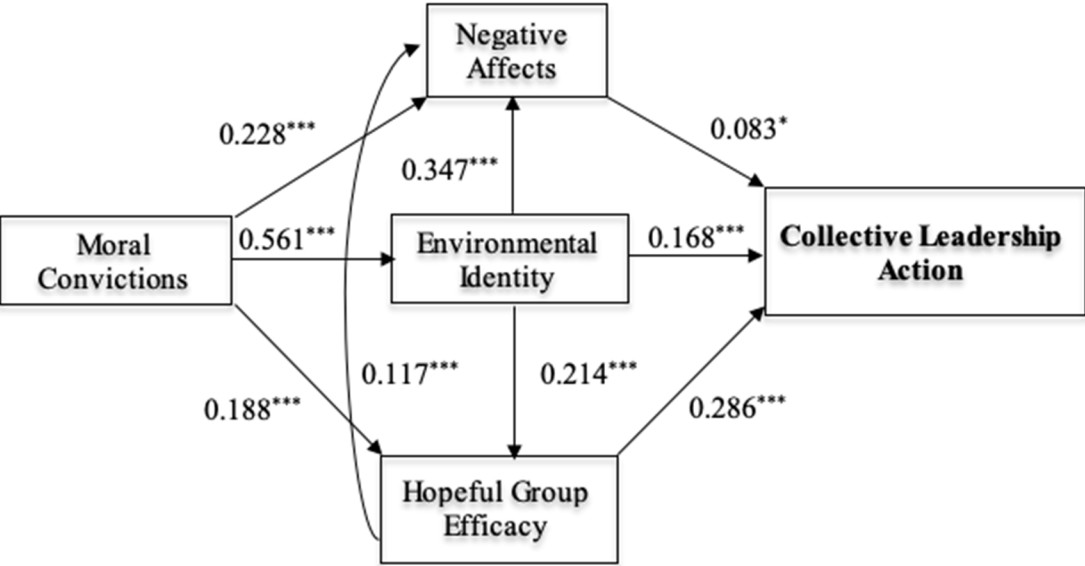

**Figure 7.** Estimates of the standardized coefficients found for the different relationship paths of the EIMECA model, including hopeful group efficacy instead of group efficacy belief: collective leadership action. * $p < 0.05$. *** $p < 0.001$.

## 12. Brief Discussion (Study 2)

The objective of Study 2 was to obtain full support for the relationships established in the EIMECA model, overcoming some of the limitations of Study 1 by improving the measurement and/or operationalization of group efficacy and the negative effects derived from perceived deterioration and environmental problems. Initially, we explored whether the inclusion of the new measure of negative affect in the model, while still maintaining the measure of group efficacy belief alone, improved the relationship between negative affect and environmental collective action measures. The estimates of the coefficients for each path were, on this occasion, all significant, including the relationship between negative affects and environmental collective action measures, although the relationship between group efficacy and these actions, as in Study 1, continued to be negative.

Therefore, the results replicated those obtained in Study 1, in terms of the positive relationships already found, and added a positive and significant relationship between negative affects and environmental collective action measures (as opposed to the absence of a relationship found in Study 1, using anger alone). This shows that when explaining

environmental collective action, consideration of the diversity of negative affects that can be derived from the perception, and concern about the deterioration of the environment and the problems that surround it, allows it to be better predicted than when using anger alone. The results, therefore, confirmed the expected effect of the negative affects on behavior set out in Hypothesis 2c of Study 2.

However, it should be noted that the relationship between group efficacy belief and environmental collective action measures was, as in Study 1, negative. Thus, the greater the group efficacy belief, the less the participation in these actions. Therefore, the positive relationship expected in Study 1 between group efficacy beliefs and environmental collective action measures was also not supported in the preliminary analyses of Study 2. However, since in Study 1 the expected effects of group efficacy belief were not obtained, in this second study our interest was focused on testing the effects of an interaction between group efficacy belief and hope. However, the results did not allow us to confirm these interactive effects, and therefore did not support Hypothesis 2c of Study 2. Given that the complementary analyses carried out suggested that rather than an interactive effect, there could be an additive effect, a new variable was constructed which we called hopeful group efficacy beliefs, which brought together the requirements of a continuous additive variable in which the high (or low) scores in both variables were added. The results of the model with this new variable revealed coefficients between all positive and significant paths. Therefore, it appears that the new variable of hopeful group efficacy beliefs explains the environmental collective action measures. Thus, according to these results, the probability of people engaging in environmental collective actions increases significantly if both variables, that is, group efficacy beliefs and hope, are high.

The model analyzed all three environmental collective action measures, including this new variable, and its relationship with both actions and negative affects yielded excellent goodness-of-fit indices. The percentage of variance explained by the model for each environmental action measurement ranged from low (17.7% for leadership action) to medium (31.2% for participation action). Environmental identity again emerged as the central variable of the model, since it obtained the highest coefficients when predicting environmental collective actions measurements, as well as being the variable that explained the most variance (31.5%).

In summary, the results of Study 2 appeared to provide further support for the relationships established in the EIMECA model, although, more than an interactive effect between group efficacy and hope, it is necessary to think about an additive effect. Most importantly, the results show that Clayton's [23] concept of environmental identity is able to successfully predict not only global collective actions, but also differentiated participation and leadership actions, being the central variable of the model due to its ability to predict, above all others, environmental collective actions both directly and indirectly, and to explain a higher percentage of variance than the rest of the variables in the model.

## 13. General Discussion and Conclusions

The main objective of the studies in this paper was to test the role of environmental identity, as conceived by Clayton [23], as a form of social—collective—identity when predicting environmental collective action, having framed the latter within the theoretical framework of the SIMCA [19] model, which then gave rise to the model that we have called EIMECA. Therefore, it was of central interest to confirm the predictive capacity of environmental identity, considering the relationships between the variables established in the model. To this end, two studies were conducted.

Study 1 confirmed the central role of environmental identity in predicting environmental collective action measures (Hypothesis 1d), thus adequately addressing the main objective of our study. It was also confirmed that moral convictions directly predict anger, group efficacy, and environmental identity (Hypothesis 1a), and that environmental identity predicts anger and group efficacy (Hypothesis 1b). With regard to Hypothesis 1c, it was only confirmed that environmental identity predicts environmental collective action

measures. Anger, as a measure of the negative affects derived from the perception of deterioration and environmental problems, failed to explain the environmental collective actions. Moreover, whilst group efficacy beliefs were able to explain such actions, this effect was in the opposite direction to that predicted by our hypothesis, that is, negative.

In Study 2, further support was sought for the relationships within the model, by addressing the potential limitations of conceptualization and operationalization of group efficacy and the negative affects of Study 1. The results initially replicated those obtained in Study 1, including the central role of environmental identity, but also the negative effects of group efficacy on environmental collective action measures. However, on this occasion, the absence of a relationship between negative affects (operationalized in Study 1 through anger) and actions did not emerge. In this second study, this relationship emerged as positive and significant, by operationalizing negative affects with a measure that allows for the assessment of a greater diversity of these affects. In relation to the negative effects of group efficacy beliefs, we found no support for an interaction between these effects and hope. However, support was obtained for an additive and simultaneous effect of high (or low) group efficacy beliefs and hope scores, that is, hopeful group efficacy belief in our study. Therefore, our study appears to highlight the important role of emotional affects, not only negative, but also positive (hope), in explaining environmental collective action.

With regard to negative affects, it is evident that at least in the domain of environmental collective action, a wide range of negative emotions better predicts behavior than anger alone. This supports the idea that when trying to explain environmental collective action, one should not only consider the negative emotions (anger, indignation, rage) that derive from the evaluation of the behaviors of others, that is, of the majority groups or groups in power, but also those that derive from the self-evaluation of one's own behavior (e.g., guilt, shame) [29,77,78].

With respect to group efficacy, we did not gain support for Cohen-Chena and van Zomeren's [76] idea that beliefs in group efficacy only motivate collective action when hope is high, but not when hope is low. However, the results revealed the important influence of high hope levels when group efficacy is high, that is, high group efficacy only has a positive effect on environmental collective action measures when it is also accompanied by high hope. Similarly, and no less importantly, it was found that these additive effects of hopeful group efficacy belief have a positive influence on the negative affects, exerting not only a direct effect on environmental collective action measures, but also an indirect effect through the negative affects. For all these reasons, it can be inferred that hope, as a positive emotion, plays a fundamental role in the decision to actively participate in collective action. Support for the additive effects obtained with respect to the hopeful group efficacy belief can be found in the results of other studies, showing that positive affect builds psychological resources such as self-efficacy, and that it promotes the commitment of individuals to the environment [83–85]. In the same vein, Aspinwall [86] concludes that positive affect influences people's assessments of the strength or adequacy of their resources for resisting negative events and information. Furthermore, the study by Coelho et al. [87] reveals that positive affect is positively related to a person's perceived self-efficacy of their environmental performance, as well as their pro-environmental behavior. Moreover, in this last study, following the proposal of Aspinwall [86], the authors suggest that these relationships reveal that people who have a high positive affect do not avoid negative information (e.g., deterioration, destruction) about the environment as a strategy to protect their feelings, but that these affects cause them to pay attention to such adverse information and to act on it, using psychological resources such as self-efficacy and adopting behaviors that protect the environment. This would therefore explain why the inclusion in our model of a positive and significant relationship between expected group efficacy and negative affects improved the fit of the model.

Further, it appears that our findings support the idea that environmental identity, as conceptualized by Clayton [23], emerges as the central axis in the EIMECA model when predicting environmental collective actions. Furthermore, this prediction was confirmed

for both global collective action measurement and for the two distinct dimensions of participation and leadership. Therefore, it can be said that both the model and the environmental identity predict participation in individually organized activities, but with the purpose of collective mobilization in favor of the environment, thereby incorporating a leadership component, as well as participation in collective activities organized by others, for the defense of nature or to increase the degree of environmental awareness, thereby incorporating a purely participatory component in environmental collective actions.

In this regard, we believe it is important to emphasize that the conceptualization of the environmental identity is in line with the concept of social-collective identity that informs and underpins current studies on collective action from a psychosocial perspective. However, the conceptualization of environmental identity offered by Clayton [23] is not a politicized identity, although it could form the basis for developing such an identity. This point is very significant, since our results on environmental identity do not support the proposal of van Zomeren et al. [28] on the unique predictive capacity of politicized identity versus collective—group identity. This is notable, because following the studies by van Zomeren et al. [19,28], the few existing studies often tend to directly assess politicized environmental identity [8,12,16,20,21], stating that group identification may not be sufficient to motivate participation in collective action. However, our results show that moral convictions can drive environmental collective action through their possible normative adjustment to the content of a collective identity—group or non-politicized, that is, environmental identity. Therefore, we believe that discarding the analysis of the relationship between non-politicized collective identity in the field of environmental collective action could be problematic. This is because we would be neglecting an alternative explanation of this behavior in those cases where there is no politicized organization associated with or representing the environmental interests of certain social groups that are still willing to participate in environmental collective action, motivated, for example, by their own identification with nature and the moral convictions related to such an identity. Thus, we understand that environmental collective action can often be a moral standard for people who are not inherently connected to politicized social-environmental movements, something that can often occur in the domain of environmental collective action. For example, as some studies reveal, there are negative stereotypes about outgoing heads or leaders of environmental groups that hinder people from joining the politicized group [88]. Moreover, although environmental social groups share common interests, membership of politicized groups representing environmental interests could vary depending on the type of collective action they tend to take (e.g., violent-aggressive versus nonviolent-non-aggressive) [14]. In short, we think that if researchers insist on considering only politicized groups in the specific field of environmental behavior, this could lead to the assumption that participation of an individual in collective actions of a group are not considered as such if they decide to participate on their own (even if this is motivated by shared interests), because such actions do not adhere to or identify with a politicized environmental group.

In short, our testing of the model has revealed the need to adapt the conceptualization and operation of various constructs of the preliminary model when explaining environmental collective action, thereby giving rise to a new proposal through the EIMECA model. From our standpoint, we believe that these adaptations derive from the distinction that needs to be made between collective actions in the field of social protest for the environment (collective action by conversion [22]) and collective actions in the field of social protest against socio-structural injustices (competitive collective action [22]). Given the results of our study, the differences between these types of collective action lead us to conclude that the models to be tested are either specific to the environmental domain and already consider these differences or must be adapted to the specific field in which environmental collective action takes place.

All this highlights the important contribution of this study to the existing literature on environmental collective action. That is, the conclusions that can be drawn from the results reflect the relevance of the present study, which contributes to existing knowledge in the

specific field of environmental collective action through the uniqueness of the EIMECA model compared with the SIMCA model. The EIMECA model, therefore, highlights the psychosocial factors that are important in predicting collective action in the specific field of social protest for the environment. The EIMECA model, apart from assigning an important role to variables proposed by the SIMCA model, such as moral convictions and group efficacy, is singularly configured by granting a central role to environmental identity when it comes to explaining environmental collective action, as opposed to the politicized identity proposed by SIMCA (without implying that a politicized environmental identity is not capable of predicting environmental collective action). Likewise, the EIMECA model is characterized by assigning greater importance to the diversity of negative effects that can emerge in response to environmental deterioration and problems (and not only anger, rage or indignation). Finally, the EIMECA model is characterized by proposing an additive measurement of group efficacy and positive emotions, such as hope. Without a strong feeling that change in environmental issues is possible, the belief that the group gathers the necessary resources to achieve change does not make participation in environmental collective actions possible. Therefore, when explaining environmental collective actions, according to the results obtained, emotions — both negative and positive in relation to the environment — play a fundamental role.

We do not want to finish this work without pointing out some potential limitations that could, to a certain extent, shape the conclusions that can be drawn from our findings. First, this is a correlational study, so it is not possible to make causal inferences regarding the direction of the relationships found. Second, it should be taken into account that our samples are composed of participants under 30 years of age and of Spanish nationality, most of whom are women. Therefore, it should be noted that the conclusions of the present study should be taken with caution in their generalization to other samples with characteristics different from the present sample. Testing the model for gender and age would be highly necessary given the absence of studies analyzing both variables in relation to environmental collective action [88]. Therefore, it is necessary to replicate the study with more heterogeneous samples, not only in terms of age and gender, but also in terms of other cultures. It would also be interesting to test the model using environmental versus non-activist samples. This would allow for greater generalization of the results found. It would also be interesting to test the model by considering the role that positive emotions can play in a wider range of emotions, in addition to hope.

Finally, it should be noted that the variance percentages obtained range between 17% and 31%. However, in the various analyzes carried out, for example, by van Zomeren et al. [19], the authors obtain variance percentages that range between 15% and 35%. According to the authors, when drawing conclusions about these results, we cannot forget the relevance of socio-structural factors and systemic influences, nor the intergroup dynamics of social conflicts. However, this is not an obstacle to conclude that the results show that subjective-psychological variables seem to be crucial when explaining collective action. These variables should be understood as psychological mediators of the factors, influences and dynamics mentioned.

## 14. Practical Implications

This study not only makes a theoretical contribution, but our results also have practical implications that are of considerable interest, at least for environmental education professionals, environmental groups, political organizations of social movements in the environmental field, and even political leaders. Thus, for example, it is worth mentioning that knowing the factors that influence the environmental behavior of young people seems relevant in any case, for educational and civic engagement purposes.

On the other hand, as suggested above, research conducted so far on the role of social identity in environmental collective action has focused on the influence of politicized environmental identity on participation in environmental collective action. This identity supposes the adhesion to particular social groups which, due to the inter-group dimension

of environmental problems, leads to a division of positions when faced with these problems and, consequently, to a confrontation and conflict with other social groups. This inter-group conflict, although unavoidable in the face of any social change, paralyzes the resolution of environmental problems [22]. One strategy for reducing inter-group conflict from a social identity perspective could be the creation of a higher-order identity that includes conflicting subgroup identities and allows for the transformation of the group context from "us" to "them" to "us" [89–92]. The environmental identity construct proposed by Clayton [23], as opposed to a politicized environmental identity, has the advantage of being composed of various dimensions of the collective environmental identity, since, in addition to the group environmental identity, it embraces, for example, identification with nature, an identity that may be common to many people regardless of their identification with many other social groups. The environmental identity of Clayton represents a good starting point for achieving this higher-order identity, and contributes toward progress in solving environmental problems. In this regard, a valuable line of future research could be to address the particular effects of each dimension of Clayton's concept of environmental identity on environmental collective action.

Furthermore, it is important to remember that, in line with the results of other studies [74], the present findings revealed rather pessimistic or hopeless feelings regarding environmental problems. Given that the results of this study also suggest the important role played by positive affects such as hope when participating in environmental collective actions, then the acquisition of such affects in the face of future environmental problems should be a fundamental aim when developing environmental education programs. The implicit message conveyed by the results of this study is that "together we can solve environmental problems" (group efficacy beliefs), "because change is possible" (hope).

**Author Contributions:** The three authors have contributed equally in all matters concerning this paper. All authors have read and agreed to the published version of the manuscript.

**Funding:** This research has been funded by the Project "Environmental Identity Model of Environmental Collective Actions (EIMECA)", granted in the call for Pre-Competitive Research Programs for Young Researchers of the Independent Research and Transfer Plan of the University of Granada (Spain), reference (PPJIB2018-04), and by the "VALCREAC" research group (HUM-196 of the Andalusian research plan of the Andalusian government).

**Institutional Review Board Statement:** This research is approved by the Research Ethics Committee of the University of Granada (Spain) with the file number on 194/CEIH/2016.

**Informed Consent Statement:** Informed consent was obtained from all subjects involved in the study.

**Data Availability Statement:** The data of this study are available on request.

**Acknowledgments:** Thanks to José Mª Salinas Martínez-de-Lecea (Department of Behavioral Sciences Methodology at the University of Granada) for his contributions and advice on methodological issues. Equally, thanks to the editors and reviewers for their help in improving this paper.

**Conflicts of Interest:** The authors declare no conflict of interest.

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
