# Peer review of "EIMECA: A Proposal for a Model of Environmental Collective Action"

_sustainability, doi:10.3390/su13115935_

Round 1
Reviewer 1 Report
See the attached document

Author Response
Reviewer 1
EIMECA: a proposal for a model of collective environmental actions
This paper represents an approach to the study of collective action from the environmental perspective. Taking as a point of reference research about collective action from Social Identity Theory as a central variable (Social Identity Model of Collective Action), the present research incorporates environmental identity as a key variable to adapt this model to the context of the environmental collective action. As a result, a theoretical proposal of environmental collective action is analyzed: The Environmental Identity Model of Environmental Collective Action (EIMECA).
Environmental Psychology has been mainly focused on analyzing proenvironmental behavior from “private-sphere environmentalism” (Stern, 2000). However, collective action or “environmental activism” -continuing with Paul Stern's classification- has been shown as one of the most remarkable actions in the 21st century with youth as the protagonist. Therefore, this work represents an important contribution to the field because of its relevance and novelty. It is also a very interesting and suggestive starting point. However, some suggestions are provided below that would help to improve the present paper.
ABSTRACT:
- Some information about methodology mut be provided.
R: See the abstract Page 1
- Acronyms should be avoided or at least their meaning should be expanded.
R: See the abstract Page 1
INTRODUCTION
In the last paragraph a reference is made to the general objectives of the study. It is mentioned that these will be achieved through two studies, but the objectives of each of them are not clear and, although the two studies are mentioned in detail later, they should be described here in general terms.
R: As the reviewer suggests the objectives of each study have been written and included in the introduction section. See page 3
CONCEPTUALIZATION OF ENVIRONMENTAL IDENTITY IN THE EIMECA MODEL
In general, the wording in section 3 is confusing. For instance, it is not clear to which author, Clayton (2003) or Tam (2013), the aforementioned results are attributed.
R: Due to changes made in response to another suggestion for improvement, the entire paragraph has been improved. See page 4.
On the other hand and about the dimensions included in the Clayton’s scale, it would be clarifying if the description is accompanied by an example of a specific item that illustrates the definition or representative meaning of such dimensions.
R: Although the reviewer's suggestion is interesting, we believe that it is not necessary to include examples of items here, since some are offered in the methodology section, thus preventing the article is unnecessarily extended. We think that, maybe, what could include in that paragraph is: "see examples of items in the methodology section." See page. 5
It would be worth reflecting on other measures of environmental identity (EID) or related concepts (connectivity, emotional affinity, ecocentrism ...) about its dimensionality and differences with Clayton's one on socio-collective identity. Although Tam’s (2013) work is referenced on this point, a more explicit reflection on what specific elements of Tajfel's Social Identity Theory could be extrapolated to Clayton's concept of EID, is lacking. On this respect I highly recommend the Doctoral Thesis by Sevillano (2006).
R: Following the reviewer's recommendation, a reflection has been prepared on other environmental identity measures or related concepts, differentiating them from the Clayton environmental identity construction and emphasizing its multidimensional construction against them. For this, according to the reviewer, sevillano's doctoral thesis (2006) has been taken into account, which has been useful as to what the theory of social identity of Tajfel refers. See page 4.
This reflection would help to better understand the following section, where the proposed model is explained.
EIMECA MODEL (ENVIRONMENTAL IDENTITY MODEL OF ENVIRONMENTAL COLLECTIVE ACTION)
In this section (4) the proposal of the theoretical model that is the aim of this paper is presented. Here the social identity variable of the SIMCA model (Social Identity Model of Collective Action) is simply replaced by environmental identity (EID). This extrapolation should be explained in more detail, reflecting on the extent to which the variables anger, moral convictions and group efficacy are specifically related to EID and justify with the literature why these particular variables and not others are the predictors of Environmental Collective Action.
R: We thank the reviewer for this suggestion as they make us see that we have been able to transmit the logic followed to argument and found the objectives of our study. However, we have to indicate that the argumentations and foundations suggested to us, are already stated in the text. More specifically, it is said that the relationship between those variables and the EID are based on those proposed by the SIMCA model and that those relationships are expected to be confirmed, since the conceptualization of Clayton's environmental identity is based on the same theory which is based on the politized social identity proposed by the SIMCA model. Therefore, the relationships expected between the variables of the EIMECA model must be similar and based on the same theoretical arguments proposed by the SIMCA model. See page 5 (the arguments or foundations of such relationships are not exposed or detailed again, for space reasons. since the SIMCA model is already explained in another section, the reader can know such arguments by reviewing the description of the SIMCA model).
In particular and considering the references to this model by van Zomeren et al. (2008, 2012), there seems to be a certain reflection on the role of moral convictions on collective action. It would be interesting to extrapolate these reflections to the field of environmental action. For instance, considering the work by Schwartz (1977) and the activation of altruistic norms; De Young (2000) and his intrinsic satisfaction variable linked to competence; cultural differences (individualism versus collectivism) in promoting environmental action (Eom, Kim, Sherman, & Ishii, 2016); etc. This necessary reflection would support the statement of the hypotheses proposed in study 1 and would guide study 2 in a more theoretically grounded way.
R: This reflection would be very interesting, but given the objectives we pursue, it is not central in this investigation and it would extend the article too over. Like this reflection we would like to include other aspects mentioned in the study, but we have not included with views that the central ideas are not covered by excess of information. The reader interested in this debate can go to the sources indicated by the reviewer and to others that already exist in the literature. The hypotheses of study 1 are supported by the arguments made in the SIMCA model.
In this regard, a more in-depth review in this section of the effect of emotions on environmental action, such as the one carried out at the beginning of the study 2, could have avoided carrying out the analyzes that only contemplate anger as an affective variable in study 1.
R: We believe that this suggestion is not taking into account the objective of the present investigation. The objective is to put the SIMCA model to the test, that is, its premises as were formulated by its authors, including also the operationalization or evaluation they made of the model variables, exception of the politicized social identity that has been removed in our study and replaced by the environmental identity as it was conceptualized by Clayton, since this is the he variable of our main interest. This is what was pursued with study 1. Once the results of study 1 have been obtained and in the light of them, study 2 was undertaken with the purpose of improving the theoretical and empirical conceptualization of those model variables that not allow satisfactory results be obtained. Therefore, the literature was reviewed with the purpose of finding suggestions for improvement that we could implement in the testing of the model. Thus that emotions are more analyzed and / or argumented in the study 2.
STUDY 2
It would be necessary to explicitly reformulate the new hypotheses or objectives that arise as a consequence of the rethinking of study 1. This would help to understand the role of the variable “hope” in the approach of the second study. For example, the introduction to section 11.1, belonging to the results chapter, would be more effective if it were presented as an introduction to study 2.
R: Following the suggestions of the reviewer, the hypotheses of study 2 have been made explicit. As to the objectives, these are explicit also at the beginning of the study 2. See page 10.
Variables and Measuring Instruments (section 10.2). The number of items that make up the variable "hope" should be indicated.
R: Following the reviewer's initiation, the number of items used to assess hope has been included. See page 11.
Results (section 11): The positive and significant relationship between negative affective states and Hope, provided by table 2 (.294, p <.01), is noteworthy; and some comments should be expected.
R: As the reviewer has suggested, such relationship has been highlighted in the text, including comments derived from it. See page 12.
Predicting Environmental Collective Action: Structural Equation Modelling (Path Analysis) (section 11.1.)
- The three previous analyzes carried out in this section could be commented on in the text without needing to include Figures 5, 6 and 7. On this regard, a note indicating that the analyzes are available from the first author would be enough. The really relevant figures of this second study are those that include the definitive variables of the model: Figures 8, 9 and 10. So many figures are excessive and can be confusing.
R: As the reviewer has suggested, the three figures have been eliminated and the analyzes have been commented on in the text. As a result of this, the remaining figures have been re-listed. See page 13
Why is the direct effect of Moral Convictions on Global Collective Action (Figure 8) and participation (Figure 9) not included in the path analysis?
R: These analyzes do not include the relationship indicated by the reviewer because, in principle, it was not necessary to fit the model, nor did it fall within our hypotheses. In the other cases (study 1), they were included at the suggestion of the AMOS program to improve the fit of the model. This is indicated in the text of the first study.
GENERAL DISCUSSION AND CONCLUSIONS
- Regarding the observations about how positive affect builds psychological resources (page 17, second paragraph) I recommend reviewing the work by Amérigo, García and López-Santiago (2018) about the effects of inducing emotions on thought-action repertoires related to nature.
R: In accordance with this suggestion from the reviewer, we have reviewed the article of Amérigo et al. And, given the affinity of ideas, we presented in section 13. Discussion and conclusions, we have included a quote from such article. See page 16 (Quote 83).
- Considering the limitations of this work in relation to replications with more heterogeneous samples (p. 19, lines 709-710), it would be interesting to make some contrasts of the model using samples of environmental activist versus non-activists.
R: Such suggestion has been included in the discussion-conclusion section. Go to page 18.
Some other general aspects which contribute to improve this paper are:
- Some of the APA norms must be checked:
Successive citations of more than two authors and the use of "et al."
R: As the editor suggests, the text has been updated, adapting the citations to the updated version of the APA´standards (7th edition).
The significant level of confidence in correlation values (see notes of Tables 1 and 2): .01 instead of 0.01
R: Corrected. See page 7 and page11
- I think there is a mistake in p.13 (Line 460): where says “got” must say “both”
R: Corrected. See page 12.
- In general and throughout the text, the term "behaviors" should be replaced by "environmental collective action measures".
R: It has been corrected in the entire article. However, in some paragraphs the word “behaviors” or “behavior” has not been deleted, because it referred to conduct in general and it was not relevant to delete it. See changes in blue color.
- In the participants’ description in both studies, it should be stated whether they come from the general population or are university students, as the data about age and gender seem to suggest.
R: It has been added between parenthesis that the sample refers to the general population. See page 10
- The fit indicators for the three models, both from Study 1 and Study 2, should appear in a table.
R: It can be put in text or in table, but not on both sites. It was placed in text to save space.
THANKS YOU VERY MUCH!
Reviewer 2 Report
Dear authors,
Your paper investigates the impact of psychological determinants in a relevant field.
However, I have a few concerns regarding the development of the hypotheses and the methodological approach.
Introduction:
I recommend you further explain based on sound empirical evidence how collective actions contribute to the mitigation of the before mentioned environmental problems. Overall, the contribution of your study remains unclear, since previous studies have already established the proposed relationships. You state that you address limitations of previous studies (line 87): Please clarify which ones.
Background and development of hypotheses:
Hypotheses for Study 2 need to be clearly stated (line 380 ff.). First, why would you test the same hypothesis with the same procedures as in Study 1 despite the fact that they had to be rejected? The consideration of additional psychological determinants (such as negative emotions such as guilt and pessimism) needs to be explained and substantiated. Also so the concept of "hopeful group efficacy" seems to be introduced ad hoc and maybe as a result of a more exploratory type analysis. This is a major flaw in the theoretical development of the empirical investigation in Study 2.
Methods:
There is a issue with your recruiting strategy and the respective sample. This type of recruiting strategy clearly leads to several sampling biases. In particular, I would like to mention the self-selection bias (e.g. when responding to a post on social media). The average age of your sample as well as the overrepresentation of females already points towards the severe selection bias. You clearly can't generalise your results the general population. You need to narrow the scope of your study to young females.
Results:
Study 1 provides no interesting contribution to research. I recommend to completely omit this study. In general, I would expect reports of a CFA or EFA for the reliability of your measures.
Discussion:
Overall, the contribution of your study remains rather limited. The overall paper might benefit from a clearer focus (in line with the sample of the empirical investigation) and a clarification of the contribution: Adaptation of an existing model.
I hope you will find my comments and suggestions helpful to further develop this paper. I wish you all the best in the continuation of your research.
Author Response
Reviewer 2
Dear authors,
Your paper investigates the impact of psychological determinants in a relevant field.
However, I have a few concerns regarding the development of the hypotheses and the methodological approach.
Introduction:
I recommend you further explain based on sound empirical evidence how collective actions contribute to the mitigation of the before mentioned environmental problems.
R: Following the reviewer's recommendation, a paragraph has been introduced that explains how collective actions contribute to the mitigation of environmental problems, supported by solid empirical research in the literature. See page 1 and 2.
Overall, the contribution of your study remains unclear, since previous studies have already established the proposed relationships. You state that you address limitations of previous studies (line 87): Please clarify which ones.
R: As far as we know, there is no other study in which the conceptualization of Clayton's environmental identity is analyzed in relation to the variables proposed by the SIMCA model. Please, if you know any study, please let us know, indicating the references. As to the limitations, we refer mainly to the lack of studies that analyze environmental identity as a different construction from politized environmental identity, under the premises of a model that allows prediction of collective environmental actions. This is clearly stated in the introduction of the article, in addition to other limitations as the reviewer can see in the comments about it in the following suggestion for improvement. See page 2 (paragraphs in green).
Background and development of hypotheses:
Hypotheses for Study 2 need to be clearly stated (line 380 ff.).
R: The hypotheses of study 2 have been included. See page 10.
First, why would you test the same hypothesis with the same procedures as in Study 1 despite the fact that they had to be rejected? The consideration of additional psychological determinants (such as negative emotions such as guilt and pessimism) needs to be explained and substantiated. Also so the concept of "hopeful group efficacy" seems to be introduced ad hoc and maybe as a result of a more exploratory type analysis. This is a major flaw in the theoretical development of the empirical investigation in Study 2.
R: With all our respect for the reviewer, we put our astonishment and disconception before these appreciations. In study 1 it is specified that the results obtained were not satisfactory for the variables “ira” and “group effectiveness beliefs”. A reflection on the same leads us to conclude that these results obey that the operational conceptualization of such variables as evaluated by the authors of the SIMCA model, does not adjust or adapt to the context of the prediction of environmental collective actions. Said evaluations were improved considering not just one emotion (anger), but a set of negative emotions because the revised literature suggested that this measure could be more interesting for such prediction. Also, the measurement of group effectiveness was improved by associating it with hope, because the revised literature also suggested it. So this variable was not included "ad hoc". All these argumentations are reasoned and explained in the article, including the reviewed studies. We ask the reviewer to reconsider our argumentations and verify that these effectively explain your doubts in this regard.
Methods:
There is a issue with your recruiting strategy and the respective sample. This type of recruiting strategy clearly leads to several sampling biases. In particular, I would like to mention the self-selection bias (e.g. when responding to a post on social media).
R: We agree with this suggestion from the reviewer, but we would be thinking that in every investigation there is a self-selection on the part of the participants. The researchers propose the participation to different groups of possible participants and they decide whether they collaborate or not with the investigation. There are different ways to address the sampling and dome of everyone is known, the researchers are forced to choose the one that allows the circumstances in which the investigation is carried out.
The average age of your sample as well as the overrepresentation of females already points towards the severe selection bias. You clearly can't generalise your results the general population. You need to narrow the scope of your study to young females.
R: These limitations were already mentioned in the conclusion section of the article. However, we have returned them to make them even more explicit. See page 18.
Results:
Study 1 provides no interesting contribution to research. I recommend to completely omit this study. In general, I would expect reports of a CFA or EFA for the reliability of your measures.
R: Again and with all respect to the reviewer, we are again astonished, puzzled by this suggestion, because in the first study the central objective of the article is tested, that is, the model as designed and operated by authors, exception of environmental identity and environmental collective actions. Study 2 was carried out because our expectations were not satisfactorally met and an attempted to improve the faults we perceived in study 1. So we think that study 1 is essential.
Discussion:
Overall, the contribution of your study remains rather limited. The overall paper might benefit from a clearer focus (in line with the sample of the empirical investigation) and a clarification of the contribution: Adaptation of an existing model.
I hope you will find my comments and suggestions helpful to further develop this paper. I wish you all the best in the continuation of your research.
R: We are been disconnected and surprised again by this suggestion, since in the introduction section of the article (see page 2) at least three major limitations of the existing literature that we intend to be covered are developed and argumented. These state part of the important contribution of the same. These are: 1) few studies that focus on the analysis of environmental identity (EID) and environmental collective action, 2) the absence of studies that test the models generated from social psychology, for explanation in the environmental context collective action, and 3) the few studies carried out have focused on analyzing group identity and, above all, politized identity, but not the construction of environmental identity as conceived by Clayton.
On the other hand, the reviewer's suggestion regarding the need to clarify the contribution of the study indicating that it is the “adaptation of an existing model”, already appears also reflected in a paragraph of section 13. General discussion and conclusions. See on page 18 in green color. However, we have added a final sentence indicating that such adaptation is an important contribution to our study.
THANKS YOU VERY MUCH!
Reviewer 3 Report
The are two main comments for the authors:
1) one significant concern refers to the generalization of conclusions, given the sample of individuals is very limited in age and biassed towards females. This is already commented as one of the potential limitations of the study in section 13. Perhaps further references about general behavioral diferences related to age and gender could contribute to the discussion of results. On the other hand, for practical implications, knowing the factors that influence environmental behavior of youngsters seems relevant in any case, for educational and civic engagement purposes.
2) The paper does not discuss if the low (17%) to medium (31%) percentage of variance explained by the model justifies the conclusions. As well as for the variance explained by the Environmental Identity factor.
Author Response
Reviewer 3
The are two main comments for the authors:
1) one significant concern refers to the generalization of conclusions, given the sample of individuals is very limited in age and biassed towards females. This is already commented as one of the potential limitations of the study in section 13. Perhaps further references about general behavioral diferences related to age and gender could contribute to the discussion of results.
R: Following the suggestion of the reviewer, an allusion to the importance of carrying out studies that consider the differences in gender and age has been included, as indicated by the first author of this article, in his recent doctorate tesis, which carried out an extensive review on these variables and the environmental collective action. See section 13. Discussion and conclusions, page 18.
On the other hand, for practical implications, knowing the factors that influence environmental behavior of youngsters seems relevant in any case, for educational and civic engagement purposes.
R: Such suggestion has been included in the practical implications section. See page 18.
2) The paper does not discuss if the low (17%) to medium (31%) percentage of variance explained by the model justifies the conclusions. As well as for the variance explained by the Environmental Identity factor.
R: As suggested by the reviewer, a paragraph has been included in section 13, in limitations of the study, in which the percentages of explained variance are discussed. See page 18
Round 2
Reviewer 2 Report
Dear authors,
Thank you for your reply. I do see a clear improvement in the conceptual development of your paper. Some of your explanations clarify the scope and contribution of your paper.
With a more detailed reading of the text a noticed a further option to improve your paper: In the empirical investigation you differentiate between a leadership and a participation dimension for collective action. However, this differentiation is not further explained in the theoretical development and the development of hypotheses. Further, examples of the measures of the different dimensions would be helpful. In addition, I miss a clear discussion of the differences and similarities of the different models (global, leadership and participation action).
However, I am afraid I still don't find the arguments regarding the development of Study 2. From my perspective, Study 1 functions as pre-test that revealed problems within your measures and conceptualisation. Therefore, the contribution for your model stems only from Study 2.
Also, I do not agree with your arguments regarding the sampling bias. A least you could test - as you propose - for the effects of gender and age.
All the best!
Author Response
Dear authors,
Thank you for your reply. I do see a clear improvement in the conceptual development of your paper. Some of your explanations clarify the scope and contribution of your paper.
- With a more detailed reading of the text a noticed a further option to improve your paper: In the empirical investigation you differentiate between a leadership and a participation dimension for collective action.
- However, this differentiation is not further explained in the theoretical development and the development of hypotheses.
- As the reviewer suggests, a large paragraph has been included in the introductory section (see page 5 in red) that addresses the theoretical development of the two dimensions of environmental collective action considered in our study. Likewise, a justification for the consideration of both dimensions and our expectations in this regard has been included.
- The allusion to the three measures of environmental collective action has been specified in the hypotheses (see pages 5 and 10 in blue).
- Further, examples of the measures of the different dimensions would be helpful.
- Following the reviewer's suggestion, examples of items of both dimensions have been included (see page 7 in red).
- In addition, I miss a clear discussion of the differences and similarities of the different models (global, leadership and participation action).
- The discussion referred to by the reviewer already appears in the manuscript. Given that the results obtained for the three collective action measures are comparable (see pages 15 and 17 in red), it is only possible to speak of the similarities, which is what is already said in these pages. However, the predictive capacity of the model has been emphasized with respect to the two dimensions of the environmental collective action measure used (see page 17 in green).
- However, I am afraid I still don't find the arguments regarding the development of Study 2. From my perspective, Study 1 functions as pre-test that revealed problems within your measures and conceptualisation. Therefore, the contribution for your model stems only from Study 2.
- The arguments that support the development of study 2 can be seen on page 10 (in green), both at the beginning of page 10, and in the two paragraphs on the same page that constitute the section: "9. Study 2".
- Study 2 was carried out due to the lack of confirmation of hypothesis 1.3 from Study 1 (see page 9 in red).
- Study 1 represents the main objective that we set ourselves at the beginning of the investigation (see pages 2 in red and 5 - “5. Study 1” - in green). We believe it necessary to highlight here that in synthesis, the main objective (as already stated in the manuscript) is to test the role of environmental identity (conceptualized by Clayton), taking into account the premises and variables proposed by the authors of the SIMCA model. In this model, the authors propose as variables: anger based on the group, the perception of group efficacy, and moral convictions (in addition to the politicized identity that in our study was proposed to be replaced by environmental identity). These, therefore, are the variables taken in our study 1, with the same approach, in terms of theoretical conceptualization and operationalization, as the authors of the model.
- Page 2: “Our main objective was to test the role of environmental identity in predicting these actions. To achieve this objective, we will take as a theoretical frame of reference one of the most relevant models on collective action that has been successfully tested in the context of socio-structural injustices, that is, SIMCA (Social Identity Model of Collective Action) [19, 20]. Likewise, we will take the conceptualization of the "environmental identity" of Clayton [23], which is the most advanced concept and also the one that most closely resembles the concept of social-collective identity from all of those existing in the environmental field [12, 26, 27]. The objective of the first study was to analyze the propositions of the SIMCA model using Clayton's environmental identity as the central axis, instead of the politicized social identity proposed by van Zomeren et al [19, 20].”
- Actually, we could have made the manuscript with just the results of study 1. Study 1 covers the main objective of our research. The results, although they did not confirm all the hypotheses (hypothesis 1.3), cover this objective. However, seeing the possibility of improving the testing of the model's premises, it was decided to carry out study 2, as already stated on pages 2-3 and 10 (in green).
- Given the main objective of our research (see page 2 in red), study 2 must have carried out study 1 first, since study 2 tries to improve the results of study 1. Without study 1, It would not make sense to carry out study 2, GIVEN THE MAIN OBJECTIVE OF OUR STUDY, among other reasons, because the deficiencies of the SIMCA model would not be known when applied in the prediction of environmental collective action.
- Also, I do not agree with your arguments regarding the sampling bias. A least you could test - as you propose - for the effects of gender and age.
- Taking into account this suggestion of the reviewer, the correlations of the two sociodemographic variables, age and gender, with the others, have been added in Tables 1 (study 1) and 2 (study 2), and it has been highlighted in the text description of results, the correlations obtained in this regard, from study 2 (see page 12 in red). Likewise, multiple linear regression analyzesper block have been carried out, in which the effects of both variables on the measures of environmental collective action were controlled. The predictor variables of the model were taken as independent variables. The results obtained have revealed that, when the effects of these sociodemographic variables are controlled (separately and jointly), all the predictor variables of the model significantly predict the three measures of environmental collective action. Therefore, and following the reviewer's suggestion, it has been verified whether the possible effects of both sociodemographic variables could distort or skew the results obtained with the path analysis (see page 13 in red).
- Regarding gender, we would like to point out that in other studies samples with a gender composition similar to ours are also obtained. For example, Schmitt et al., 2019, N = 152; men = 49; women = 103. Another example, Dono et al., 2010, N = 231; men = 27; women = 104). Although this does not occur in other studies, it is possible that these samples, like ours (which were independent samples, collected at two different time points), are somehow reflecting the real tendency of women and men when it comes to decide to participate in environmental studies. This data has been extremely interesting to us and we believe that it could be the object of study in the future.
- In relation to the suggestions for improvement marked by the reviewer in the “Review Report Form”:
- The entire manuscript has been reviewed for the minor spelling check suggested by the reviewer and some of the corrections have been marked in red.
- Regarding the rest of these suggestions, we hope that with the modifications made to the manuscript in response to the reviewer's indications, we have managed to improve: the theoretical background and empirical research on the subject (see pages 2, 5) research hypotheses and methods (see pages 5, 7, 9, 10, 12, 13), arguments and discussion of findings (see pages 15, 17), presentation of results (see pages 7, 11 , 13), and references (see pages 20-23).